# Integrating a host biomarker with a large language model for diagnosis of lower respiratory tract infection

Hoang Van Phan[1,6], Natasha Spottiswoode [1,6] ✉, Emily C. Lydon[1], Victoria T. Chu [2,3], Adolfo Cuesta[1], Alexander D. Kazberouk[4], Natalie L. Richmond[1], Padmini Deosthale[1], Carolyn S. Calfee[5] & Charles R. Langelier [1,3]

Lower respiratory tract infections (LRTI) are a leading cause of mortality and are challenging to diagnose in critically ill patients, as non-infectious causes of respiratory failure can present with similar clinical features. We develop an LRTI diagnostic method combining the pulmonary transcriptomic biomarker *FABP4* with electronic medical record text assessment using the large language model Generative Pre-trained Transformer 4. In a cohort of critically ill adults, a combined classifier incorporating *FABP4* expression and large language model electronic medical record analysis achieves an area under the receiver operating characteristic curve (AUC) of 0.93 ± 0.08 and an accuracy of 84%, outperforming *FABP4* expression alone (0.84 ± 0.11) and large language model-based analysis alone (0.83 ± 0.07). By comparison, the medical team admission diagnosis has an accuracy of 72%. In an independent validation cohort, the combined classifier yields an AUC of 0.98 ± 0.04 and accuracy of 96%. This study suggests that integrating a host biomarker with large language model analysis can improve LRTI diagnosis in critically ill adults.

Lower respiratory tract infections (LRTI) are a leading cause of death worldwide, yet remain challenging to diagnose[1]. This is especially true in the intensive care unit (ICU), where non-infectious acute respiratory illnesses often have similar clinical manifestations. Further complicating accurate diagnosis is the failure to identify a causative pathogen in most clinically recognized cases of LRTI[2]. The resulting diagnostic uncertainty drives the overuse of empiric antibiotics, leading to adverse outcomes ranging from *Clostridioides difficile* infection to the development of antimicrobial resistance[3,4].

Host transcriptional biomarkers are a promising modality for LRTI diagnosis that overcome several limitations of traditional microbiologic tests[5,6]. By offering a more direct and dynamic measure of the

host immune response, they can enable earlier and more accurate identification of infection and can differentiate bacterial and viral etiologies, even in cases where pathogen detection is unsuccessful[6,7]. Single gene biomarkers are particularly amenable to clinical translation, as they can be readily incorporated into simple nucleic acid amplification platforms which are already widely used in healthcare settings.

The pulmonary expression of the gene *FABP4*, for instance, was recently identified as an LRTI diagnostic biomarker in critically ill patients with acute respiratory failure, achieving an area under the receiver operating characteristic curve (AUC) of 0.85 ± 0.12 in adults and 0.90 ± 0.07 in children[8]. *FABP4*'s consistent performance across

[1]Department of Medicine, Division of Infectious Diseases, University of California San Francisco, San Francisco, CA, USA. [2]Department of Pediatrics, Division of Infectious Diseases and Global Health, University of California San Francisco, San Francisco, CA, USA. [3]Chan Zuckerberg Biohub San Francisco, San Francisco, CA, USA. [4]Department of Medicine, University of California San Francisco, San Francisco, CA, USA. [5]Department of Medicine, Division of Pulmonary, Critical Care, Allergy and Sleep Medicine, University of California San Francisco, San Francisco, CA, USA. [6]These authors contributed equally: Hoang Van Phan, Natasha Spottiswoode. ✉e-mail: natasha.spottiswoode@ucsf.edu

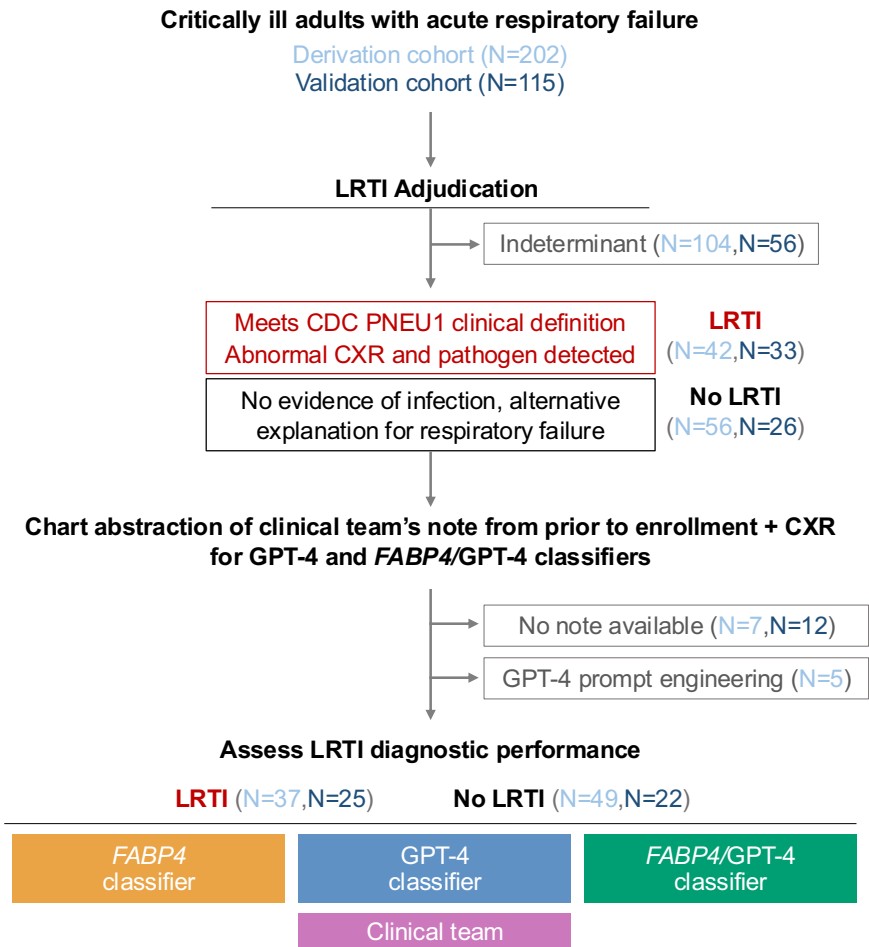

**Fig. 1 | Study flow diagram and overview of the derivation and validation cohorts.** Abbreviations: *LRTI* lower respiratory tract infection; *RNA-seq* RNA sequencing; *CXR* = chest X ray, *FABP4* gene encoding fatty acid binding protein 4; *CDC* U.S. Centers for Disease Control and Prevention; *GPT-4* Generative Pre-trained Transformer 4.

cohorts with differing microbiology–predominantly bacterial infections in adults and viral in children–suggested that *FABP4* is agnostic to the type of pathogen causing LRTI[8]. Mechanistically, fatty acid binding proteins are a family of intracellular proteins that modulate fatty acid trafficking and inflammatory signaling[9]. *FABP4* specifically is highly expressed in tissue-resident alveolar macrophages which are preferentially depleted during respiratory infections including bacterial pneumonia[10] and COVID-19[11]. While its performance as a LRTI biomarker exceeds clinical biomarkers such as C-reactive protein[12] or procalcitonin[13], *FABP4* alone likely does not achieve the accuracy necessary to enable confident clinical decisions regarding antimicrobial use in critically ill patients with acute respiratory failure.

Large language models (LLMs) such as Generative Pre-trained Transformer 4 (GPT-4) represent a new class of artificial intelligence tools with potential utility across a diversity of medical applications[14]. GPT-4 provides a text interface in which a clinician or other user may pose questions, to which GPT-4 then responds in conversational language. While LLMs have demonstrated remarkable performance for some medical use cases, including image interpretation[15–17] and patient risk stratification[18], their utility in aiding clinical reasoning remains unclear[19–21], and their potential role for diagnosing LRTI or other critical illness syndromes based on electronic medical record (EMR) data has not been assessed. Furthermore, evaluation of LLMs in combination with other tools, such as host biomarkers, remains largely unexplored.

Here, we address this gap by building a diagnostic classifier combining *FABP4* expression with GPT-4 analysis of electronic medical record (EMR) data. We find that this combination affords remarkably

accurate LRTI diagnosis, suggesting a promising approach to improve the care of critically ill patients.

## Results

We evaluated the performance of four different diagnostic approaches (*FABP4*, GPT-4, integrated *FABP4*/GPT-4 classifier, and admission diagnosis by the primary medical team) against a gold-standard of retrospective LRTI adjudication performed by two or more physicians. The derivation cohort included 42 patients with LRTI and 56 with no evidence of infection and a clear alternative explanation for respiratory failure, and the external validation cohort included 33 LRTI cases and 26 No LRTI cases (Fig. 1). The majority of LRTI patients in the derivation cohort had bacterial etiologies of LRTI, while the validation cohort, which was largely recruited during the COVID-19 pandemic, was predominantly viral (Table 1). Time to microbiologic diagnosis from intubation was a median of 56.3 h (interquartile range (IQR) 28.9–73.4 h) in the derivation cohort and 59.6 (33.9–88.8 h) in the validation cohort for patients with diagnoses other than COVID-19. Due to the incorporation of universal SARS-CoV-2 testing prior to hospital admission during the period of validation cohort enrollment, most patients with COVID-19 were diagnosed prior to intubation.

We provided GPT-4 with practical clinical summary information from the EMR that would be available to a treating physician assuming care of a patient in the ICU: a chest x-ray (CXR) radiology report from the day of enrollment and the note written by the medical team from the day prior. In the derivation cohort, notes and radiology reports from five patients were utilized for GPT-4 prompt engineering and optimization (**Methods**) and seven lacked a clinical note from the day

**Table 1 | Clinical and demographic features of cohorts**

| | Derivation Cohort | | | Validation Cohort | | |
|---|---|---|---|---|---|---|
| | LRTI | No LRTI | P | LRTI | No LRTI | P |
| **N** | 37 | 49 | | 25 | 22 | |
| **Age, years** (Median, Q1-Q3) | 65.0 (51.0 – 75.0) | 62.0 (53.0-73.0) | 0.81 | 53.9 (49.4 – 66.9) | 58.5 (57.5 - 63.9) | 0.69 |
| **Female Sex** (No., %) | 11 (30) | 30 (61) | 0.0074 | 12 (48) | 8 (36) | 0.61 |
| **Race** (No., %) | | | 0.91 | | | 0.10 |
| White | 18 (49) | 24 (49) | - | 5 (20) | 10 (46) | - |
| Black/African American | 4 (11) | 5 (10) | - | 2 (8) | 1 (5) | - |
| Asian | 8 (22) | 9 (18) | - | 3 (12) | 4 (18) | - |
| Native Hawaiian/Pacific Islander | 1 (3) | 0 (0) | - | 1 (4) | 2 (9) | - |
| Other/Unknown | 6 (16) | 11 (22) | - | 14 (56) | 5 (23) | - |
| **Hispanic ethnicity** (No., %) | 5 (14) | 11 (22) | 0.44 | 10 (40) | 5 (23) | 0.40 |
| **Comorbidities** (No., %) | 36 (97) | 45 (92) | 0.54 | 24 (96) | 17 (77) | 0.14 |
| **Immunosuppressed** (No., %) | 8 (22) | 6 (12) | 0.38 | 1 (4) | 3 (14) | 0.51 |
| **Microbiologic diagnosis** (No., %) | | | | | | |
| Bacterial | 28 (76) | | | 3 (12) | | |
| Viral | 4 (11) | | | 14 (56) | | |
| SARS-CoV-2 | 0 (0) | | | 13 (52) | | |
| Fungal | 1 (3) | | | 0 (0) | | |
| Multiple | 4 (11) | | | 8 (32) | | |
| **No LRTI group cause of respiratory failure** (No., %) | | | | | | |
| Surgery | | 14 (29) | | | 6 (27) | |
| Neurologic | | 12 (25) | | | 7 (32) | |
| Cardiovascular | | 8 (16) | | | 4 (18) | |
| Non-LRTI infection | | 5 (10) | | | 6 (27) | |
| Other | | 11 (22) | | | 2 (9) | |
| **ICU admission diagnosis** (No., %) | | | | | | |
| LRTI | 37 (100) | 24 (49) | - | 25 (100) | 7 (32) | - |
| No LRTI | 0 (0) | 25 (51) | - | 0 (0) | 15 (68) | - |
| **Clinical team writing note** (No., %) | | | 0.20 | | | <0.0001 |
| Internal Medicine | 15 (41) | 9 (18) | - | 1 (4) | 5 (23) | - |
| Critical Care | 5 (14) | 7 (14) | - | 24 (96) | 6 (27) | - |
| Neurosurgery | 3 (8) | 9 (18) | - | 0 (0) | 4 (18) | - |
| Cardiology | 3 (8) | 6 (12) | - | 0 (0) | 2 (9) | - |
| Other | 11 (30) | 18 (37) | - | 0 (0) | 5 (23) | - |
| **Note to enrollment** (No., %) | | | 0.080 | | | 0.11 |
| 1 day | 34 (92) | 49 (100) | - | 21 (84) | 22 (100) | - |
| 2 days | 3 (8) | 0 (0) | - | 4 (16) | 0 (0) | - |
| **CXR to enrollment** (No., %) | | | 0.24 | | | 0.19 |
| 0 days | 30 (81) | 33 (67) | - | 19 (76) | 11 (50) | - |
| 1 day | 7 (19) | 13 (27) | - | 4 (16) | 9 (41) | - |
| 2 days | 0 (0) | 3 (6) | - | 2 (8) | 2 (9) | - |
| **Hours from intubation to sample** (Median, Q1-Q3) | 26.4 (19.7 – 34.2) | 24.0 (17.5 – 49.4) | 0.82 | 33.5 (18.0 –46.2) | 23.9 (14.0 –34.4)* | 0.19 |

Chi-squared test used for all categorical variables except time from note to enrollment for which Fisher exact test was used. Two-sided Wilcoxon rank-sum test was used for continuous variables. IQR = interquartile range. One No LRTI patient in the derivation cohort, and three No LRTI patients in the validation cohort, were adjudicated as having ≥1 etiology of respiratory failure. Four LRTI patients in the derivation cohort were adjudicated as having more than one microbiologic diagnosis (three with viral-bacterial co-infection, and one with viral-fungal co-infection) and eight LRTI patients in the validation cohort were adjudicated as having more than one microbiologic diagnosis (all with SARS-CoV-2-bacterial co-infection). *Excluding 9 patients with SARS-CoV-2 intubated at other hospitals prior to transfer.

prior to study enrollment, leaving a total of 37 LRTI and 49 No LRTI cases available for analysis.

We first compared the accuracy of the primary medical team's ICU admission diagnosis, extrapolated by their decision to prescribe antimicrobials, against the gold-standard retrospective LRTI adjudication. The medical team correctly identified 37/37 (100%) of true LRTI cases but incorrectly called LRTI in 24/49 (49%) of patients in the No LRTI group, equating to an accuracy of 72% (Fig. 2A, and Table 1). All 24 of the derivation cohort No LRTI patients unnecessarily treated for LRTI

received antibacterial coverage, with four additionally receiving empiric therapy directed at viral and/or fungal pathogens (Supplementary Table 1). We next assessed the diagnostic performance of *FABP4* and found that it achieved an AUC of 0.84 ± 0.11 (mean ± standard deviation) by five-fold cross validation (Fig. 2B). We then assessed the performance of GPT-4 in diagnosing LRTI, with three independent diagnoses, resulting in a score from 0-3 for each patient. A logistic regression classifier based on this GPT-4 score achieved an AUC of 0.83 ± 0.07 (Fig. 2B).

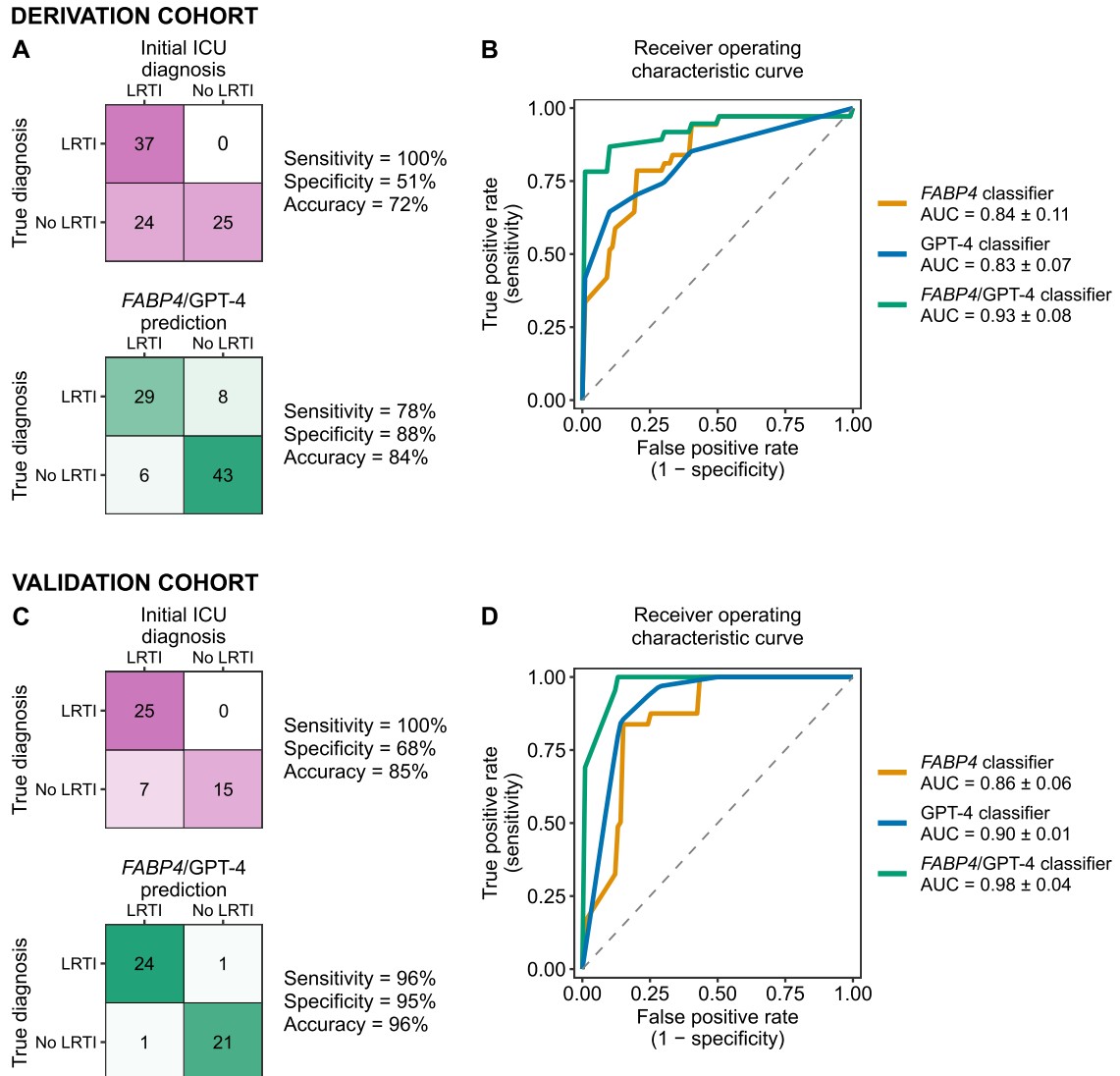

**Fig. 2 | Performance of *FABP4*, GPT-4 and integrated LRTI diagnostic classifiers in the derivation and validation cohorts. A** Confusion matrices for initial ICU diagnosis and the integrated *FABP4/*GPT-4 classifier in the derivation cohort. **B** Receiver operating characteristic curves from GPT-4 classifier, *FABP4* classifier, and integrated *FABP4/*GPT-4 classifier in the derivation cohort. **C** Confusion matrices for initial ICU diagnosis and the integrated *FABP4/*GPT-4 classifier in the validation cohort. **D** Receiver operating characteristic curves from GPT-4 classifier,

*FABP4* classifier, and integrated *FABP4/*GPT-4 classifier in the validation cohort. In (**A** and **C**), the classifiers output an LRTI diagnosis if the patients had a predicted out-of-fold LRTI probability of 50% or higher, Intensity of color in confusion matrices reflects percentage of patients in each quadrant; red indicates the initial ICU diagnosis and green is the integrated *FABP4/*GPT-4 classifier. In (**B** and **D**), the area under the curves (AUCs) are presented as mean ± standard deviation.

Combining *FABP4* and GPT-4 into a single logistic regression model achieved an AUC of 0.93 ± 0.08 (Fig. 2B), outperforming both *FABP4* (P = 0.002, one-sided paired t-test) and GPT-4 alone (P = 0.008, one-sided paired t-test). We used the one-sided paired t-test because we were primarily interested in whether the integrated classifier outperformed each individual classifier, but the difference remained statistically significant when using two-sided t-tests (P = 0.004 and P = 0.016, respectively). Considering an out-of-fold probability of 50% as LRTI-positive, the integrated *FABP4/*GPT-4 classifier had a sensitivity of 78%, specificity of 88%, and accuracy of 84% (Fig. 2A). Assessment of the integrated classifier's performance at the Youden's index within each cross-validation fold demonstrated an average sensitivity of 86%, specificity of 98%, accuracy of 93%, positive predictive value of 97%, negative predictive value of 91%, and F1 score of 0.91.

Next, we assessed the validation cohort (Fig. 1), in which the primary medical team correctly identified 25/25 (100%) of LRTI cases but unnecessarily treated with antibacterials in 7/22 (32%) of patients in the No LRTI group, equating to an accuracy of 85% (Fig. 2C, Supplementary

Table 1). The integrated *FABP4/*GPT-4 classifier achieved a sensitivity of 96%, specificity of 95%, and accuracy of 96%, again outperforming both *FABP4* alone (accuracy 79%) and GPT-4 alone (accuracy 79%). In the validation cohort, the integrated classifier achieved an AUC of 0.98 ± 0.04 using 3-fold cross-validation, as compared to *FABP4* (0.86 ± 0.06) or GPT-4 (0.90 ± 0.01) alone (P = 0.08 and P = 0.02, respectively, one-sided paired t-test) (Fig. 2D).

To gain insight into how GPT-4 returns diagnoses based on limited information, we compared the LLM against the decision making of three comparison physicians who were provided identical input. From the same limited EMR data and prompt provided to GPT-4, we asked the comparison physicians to assign a diagnosis of LRTI or No LRTI for each patient in the derivation cohort. Considering a threshold of at least one LRTI diagnosis per patient across the three physicians as LRTI-positive, we found a sensitivity of 78%, specificity of 88%, and accuracy of 84% (Fig. 3A). Finally, we sought to identify potential biases in GPT-4 diagnoses by comparing GPT-4 results to those of the comparison physicians (Fig. 3B), focusing on cases with two or more

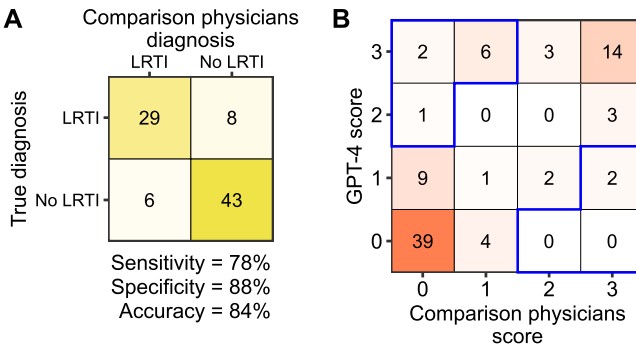

**Fig. 3 | Comparison of GPT-4 performance to physicians provided the same EMR data from the LRTI derivation cohort. A** Confusion matrix of diagnosis by three GPT-4 comparison physicians who received the same prompt and data as GPT-4. **B** Comparison of GPT-4 LRTI scores as compared to physicians. In **B** Y axis depicts the number of times GPT-4 diagnosed LRTI out of 3, X axis shows the number of times the physicians called LRTI out of 3. Blue boxes indicate instances in which GPT-4 diagnoses were most discordant with comparison physicians (the scores differ by 2 or more).

discordant LRTI diagnoses. Of the nine patients more frequently diagnosed with LRTI by GPT-4 versus the comparison physicians (Fig. 3B), six had clinical notes with no mention of LRTI, but explicit concern for LRTI in the CXR report, as judged by mention of "pneumonia" and/or "infection" in the radiologist read (CXR reports provided in **Supplementary Appendix 4**). This suggested that GPT-4 may have placed more weight on CXR reads relative to physicians. Of the two patients disproportionately diagnosed with LRTI by comparison physicians versus GPT-4 (Fig. 3B), one had a final diagnosis of e-cigarette/vaping associated lung injury, and the other had LRTI attributed to rhinovirus.

## Discussion

Our findings demonstrate that the combination of a host tran-scriptomic biomarker with AI assessment of EMR text data can improve LRTI diagnosis in critically ill patients. We found that an integrated *FABP4/*GPT-4 classifier achieved higher LRTI diagnostic accuracy than *FABP4* alone, GPT-4 alone, or the treating medical team. In our study population, we found that the initial treating physicians unnecessarily prescribed antimicrobials in a third to half of patients ultimately found to have non-infectious causes of acute respiratory failure. All patients who inappropriately received antimicrobials received antibacterial therapy, and a smaller number additionally received antiviral and/or antifungal therapy. Had our integrated clas-sifier results been theoretically available at the time of ICU admission, we estimate that inappropriate antimicrobial use might have been prevented in 20/24 (83%) and 7/7 (100%) of No LRTI patients who were unnecessarily treated in the derivation and validation cohorts, respectively. Acute respiratory illness is a leading reason for inap-propriate antibiotic use[22], and our results suggest a potential role for biomarker/AI classifiers in antimicrobial stewardship, a major goal of the U.S. CDC[23] and the World Health Organization[24]. However, given the challenges of de-escalating antimicrobials in critically ill patients, and the potential consequences of inappropriately stopping treatment in a patient with true LRTI, our results serve primarily as a proof of concept requiring further validation.

Finally, many patients with clinical LRTI never have a confirmed microbiologic diagnosis[2]; across the study cohorts, approximately 50% of enrolled patients were clinically adjudicated as having LRTI without an identified microbial pathogen, or of having an indeterminate LRTI status (Fig. 1). Although we focused on the unequivocal cases of proven LRTI or No LRTI to develop and test our GPT-4 and *FABP4/*GPT-4 classifiers, it is those cases without a clear diagnosis, in which LRTI is

considered as one possible diagnosis among many, where this method may ultimately prove most useful. A future randomized clinical trial will be needed to conclusively test this.

Previous studies have found that GPT-4 is influenced by the pre-cise language used in a prompt, leading to a need for prompt engineering[20]. By iterating our prompt on a subset of patients, and through direct comparison to physicians provided with identical EMR data, we identified possible blind spots of GPT-4 and gained insights that may help guide future optimization of LLMs for infectious disease diagnosis.

A primary strength of this study is the combination of a host transcriptional biomarker with AI interpretation of EMR text data to advance infectious disease diagnosis. We address one of the most common and challenging diagnostic dilemmas in the ICU, leverage deeply characterized cohorts, and employ a rigorous post-hoc LRTI adjudication approach incorporating multiple physicians.

Importantly, clinicians with access to a HIPAA-compliant GPT-4 interface can readily use our prompt without any prior bioinformatics expertise. Moreover, this approach yielded promising results in two cohorts of patients with very different microbial etiologies of LRTI, suggesting that both *FABP4* and LLM analysis of EMR data may have utility as diagnostic approaches, agnostic to type of LRTI pathogen. Bacterial LRTI was predominant in the pre-pandemic derivation cohort while the validation cohort, which was enrolling during the COVID-19 pandemic, primarily consisted of patients with viral LRTI.

Limitations of this study include a relatively small sample size, particularly in the validation cohort, which necessitated the use of 3-fold (versus 5-fold) cross-validation. In addition, our focus on mechanically ventilated patients may limit generalizability to less severe respiratory illnesses. Antimicrobial administration is an imper-fect proxy for clinical team LRTI diagnosis; however, it was an objec-tive, reproducible and unbiased option for retrospectively estimating the clinical team's decision making. We restricted GPT-4 analyses to a single EMR note and CXR read, and it is possible that assessment of more complete EMR data would have led to improved, or different, performance.

Given these limitations, this study is best seen as a proof-of-concept that establishes the feasibility and promise of a diagnostic approach combining artificial intelligence-based EMR analysis with a host biomarker. Future work can test whether GPT-4 can improve the marginal performance of widely available clinical biomarkers such as C-reactive protein, assess the generalizability of *FABP4/*GPT-4 classifier performance in larger independent cohorts of ICU patients, and eval-uate these methods for the diagnosis of other critical illness syn-dromes such as sepsis.

## Methods
### Adjudication of LRTI status
Gold standard adjudication of LRTI status was performed retro-spectively following ICU discharge by two or more physicians using all available information in the EMR, and based on the U.S. Centers for Disease Control and Prevention (CDC) PNEU1 criteria[27] as well as an identified pulmonary pathogen. Patients with negative microbiological testing and a clear alternative reason for their acute respiratory failure besides pulmonary infection, representing the clinically relevant con-trol group, were also identified (No LRTI group). Any adjudication discrepancies were resolved by a third physician, and patients with indeterminate LRTI status were excluded.

### Extraction of EMR data
The primary medical or ICU team's clinical note from the day prior to study enrollment and the CXR read from the day of enrollment were extracted from the EMR. If no note was written on the day prior to enrollment, a note from two days prior was substituted (Table 1). Notes were written in the EPIC EMR platform by physicians from the primary

care team, which included Internal Medicine, Critical Care, and several other additional services (Table 1). Notes varied in length and structure, reflecting the real-world diversity of clinical practice, and allowing a realistic scenario for GPT-4 use. If no CXR was performed on the day of enrollment, the next closest CXR read prior to the date of enrollment was used instead. Patients with no clinical notes available prior to study enrollment were excluded (N = 7 derivation cohort, N = 12 validation cohort). The clinical treatment team's LRTI diagnosis was extrapolated based on administration of empiric antimicrobials (antibacterial, antiviral, and/or antifungal agents) for at least 24 h within one day of study enrollment, excluding agents given for established non-pulmonary infections or prophylaxis.

## RNA sequencing

RNA was extracted from tracheal aspirates collected on the day of enrollment and underwent rRNA depletion followed by library preparation using the NEBNext Ultra II kit on a Beckman-Colter Echo liquid handling instrument, as previously described[8]. Finished libraries underwent paired-end sequencing on an Illumina NovaSeq.

## FABP4 diagnostic classifier

All analyses were done in R version 4.5.0. *FABP4* expression was normalized using the varianceStabilizingTransformation function from DESeq2 package (v1.48.1)[28], and used to train a logistic regression classifier. We chose to use logistic regression because among machine learning methods, it was best suited for the 1-2 features we sought to test within the sample size of the cohorts. More specifically, logistic regression is less vulnerable to overfitting compared to other more complex models, such as a random forest or gradient boosting classifiers. In addition, logistic regression is among the most broadly utilized statistical methods reported in the medical literature, and we believe that this inherent familiarity and interpretability would be particularly appealing in clinical settings compared to more advanced but less transparent machine learning models.

In each iteration of 5-fold cross-validation, both training and test sets were filtered to retain only genes with at least 10 counts across 20% of the samples in the training set. The test fold's *FABP4* expression level was normalized using variance-stabilizing transformation and the dispersions of the training folds, and input to the trained logistic regression classifier to assign LRTI or No LRTI status for each patient in the test fold. The performance and receiver operating characteristic (ROC) curve for each of the five folds was evaluated using the package pROC v1.19.0.1[29]. The mean AUC and standard deviation were calculated from the average AUC derived from each test fold. The sensitivity and specificity at Youden's index were extracted for each test fold separately using the function coords from the pROC package, and the average and standard deviation was calculated across the cross-validation folds.

## GPT-4 input, scoring, and prompt engineering

We used the GPT-4 turbo model with 128k context length and a temperature setting of 0.2, implemented in Versa, a University of California, San Francisco (UCSF) Health Insurance Portability and Accountability Act (HIPAA)-compliant model. For each patient, compiled clinical notes and CXR reads were input into the GPT-4 chat interface. Prompt engineering was initially carried out by iterative testing on clinical notes and CXR reads from five randomly selected patients in the derivation cohort, who were excluded from subsequent analyses. We employed a chain-of-thought prompt strategy[30] that involved asking GPT-4 to analyze the note and CXR step-by-step. The validation cohort included patients enrolled during the height of the COVID-19 pandemic and thus we redacted the terms "SARS-CoV-2" or "COVID-19" from their notes to avoid biasing the GPT-4 analysis. In our final version of the prompt (**Supplementary Appendix 1**), we asked

GPT-4 to choose either LRTI or no LRTI, as exemplified in two example responses (**Supplementary Appendix 2 and 3**). We found that GPT-4 would sometimes give different answers to the same prompt and EMR input data in separate chat sessions. Therefore, for each patient, GPT-4 was asked to diagnose LRTI in three separate sessions. A per-patient GPT-4 score was calculated based on the total number of LRTI-positive diagnoses made by GPT-4 (ranging 0-3).

## Integrated classifier

The integrated classifier's performance was tested using 5-fold cross-validation in the derivation cohort. Because of the smaller sample size, 3-fold cross-validation was used in the validation cohort. For each test fold, a logistic regression classifier was trained on the remaining training folds using both normalized *FABP4* expression and the GPT-4 score. The performance and ROC curve for each fold was evaluated as described above. The sensitivity, specificity and accuracy were calculated based on an out-of-fold predicted LRTI probability threshold of greater than or equal to 50%.

## Comparing GPT-4 to physicians provided the same data

We compared LRTI diagnosis by GPT-4 against LRTI diagnosis made by three physicians trained in internal medicine (ADK) or additionally subspecializing in infectious diseases (AC, NLR). The physicians were provided with identical information and prompts as GPT-4, and they were asked to assign each patient as either LRTI or No LRTI. The comparison physician group score (0–3) was calculated based on the total number of LRTI-positive diagnoses made by the comparison physicians.

## Ethics statement

We studied patients from two prospective observational cohorts of critically ill adults with acute respiratory failure enrolled within 72 h of intubation at the University of California San Francisco (UCSF) Medical Center (Fig. 1, Table 1). The derivation cohort[7] (N = 202) was enrolled between 10/2013 and 01/2019, and validation cohort (N = 115) was enrolled between 04/2020 and 12/2023. This research was approved by the University of California, San Francisco Institutional Review Board (IRB) under the following protocols: #10-02701 for the derivation cohort, and #20-30497 and #10-02852 for validation cohort.

If a patient met inclusion criteria, then a study coordinator or physician obtained written informed consent for enrollment from the patient or their surrogate. Patients or surrogates were provided with detailed written and verbal information about the goals of the study, data and specimens that would be collected, and potential risks to the subject. Patients and their surrogates were also informed that there would be no benefit to them from enrollment in these studies and that they may withdraw informed consent at any time during the course of the study. All questions were answered, and informed consent documented by obtaining the signature of the patient or their surrogate on the consent document or on an IRB-approved electronic equivalent. As previously described[25,26], the IRB granted an initial waiver of consent for patients who could not provide informed consent at time of enrollment.

More specifically, subjects who were unable to provide informed consent at the time of enrollment could have biological samples as well as clinical data from the medical record collected. Surrogate consent was actively pursued, and each patient was regularly examined to determine if and when they would be able to consent for themselves. For patients whose surrogates provided informed consent, direct consent from the patient was then obtained if they survived their acute illness and regained the ability to consent. A full waiver of consent was approved for subjects who died prior to consent being obtained. Further details on the enrollment and consent process for these studies can be found in two recent publications[25,26].

**Reporting summary**

Further information on research design is available in the Nature Portfolio Reporting Summary linked to this article.

## Data availability

Source data are provided with this paper. The human raw sequencing data are protected due to data privacy restrictions from the IRB protocol governing patient enrollment, which protects the release of raw sequencing data from those patients enrolled under a waiver of consent. To honor this, researchers who wish to obtain FASTQ files for the purposes of independently generating gene counts can contact the corresponding author (natasha.spottiswoode@ucsf.edu) and request to be added to the IRB protocol. All patient demographic data, sample metadata, and processed host gene counts needed to replicate this study are available in the GitHub repository, and all data generated in this study are available in the code outputs file and in the Source Data file. Source data are provided with this paper.

## Code availability

All input data and code used in this study are available at https://github.com/infectiousdisease-langelier-lab/LRTI_FABP4_GPT4_classifier and at Zenodo (10.5281/zenodo.17362824).

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

## Acknowledgements

This work was supported by the National Institution of Allergy and Infectious Diseases, National Institutes of Health [grant number R01AI185511 to C.R.L.], the National Heart, Lung, and Blood Institute, National Institutes of Health [grant number NHLBI R35HL140026 to C.S.C.], and the Chan Zuckerberg Biohub [C.R.L.].

## Author contributions

H.V.P., N.S., and C.R.L. conceived and designed the study. C.S.C., C.R.L. supervised the study. E.C.L., V.T.C., and P.D. acquired the data. A.C., A.D.K., and N.L.R. provided comparison physician diagnoses. H.V.P., N.S., C.R.L., E.C.L., and V.T.C. analyzed and interpreted the data. H.V.P., N.S., and C.R.L. wrote the manuscript. E.C.L., V.T.C., and C.S.C. reviewed the manuscript.

## Competing interests

No authors report financial or personal conflicts of interest.
