## [Peer Review File · Nature Communications]

Integrating a host biomarker with a large language model for diagnosis of lower respiratory tract infection

Corresponding Author: Dr Natasha Spottiswoode

Version 0:

Reviewer comments:

Reviewer #1

(Remarks to the Author)

I read with interest the work from Van Phan et al. In their work, the authors proposed a diagnostic system for low respiratory tract infection. The system is based on large language models where the input is electronic medical record combined with measurements of the transcriptomic biomarker FABP4. While this is a remarkable and easy-to-read paper that bridges well the language of machine learning and medical sciences, a future iteration of the work might benefit from clarification of the following points:

My biggest concern of the paper is regarding potential biases that might have been introduced in the EMRs that might have influenced GPT4's decision. The authors did not make clear the level of details of each EMR that was fed to GPT4. This needs to be considered as GPT models are sensitive to phrasing, structure, and format of input data. What are the similarities between the EMRs? Without a clear standardization, GPT may perform better on some groups.

The authors used only logistic regression models to predict (FABP4, GPT4, and both). What is the rationale on using this model? Were other models tested?

In lines 115-117, the authors bring up the classification metrics of their combined classifier. The authors could have used different metrics that consider the imbalanced nature of their adjudication data.

A clear benefit demonstrated by the authors in using such system was its ability to classify true negatives (where the medical team incorrectly classified ~70% of the true negatives as false positives). The performance in the automated system and the medical team to classify true positives was similar (96% versus 100%, respectively). In order to better justify the use of the proposed system, I suggest the authors to add insights on how long it took for the medical team to make the diagnosis of the true positive classes.

(Remarks on code availability)

The code and data are publicly available. The code is well documented and easy to reproduce.

Reviewer #2

(Remarks to the Author)

This is an interesting study that attempts to combine a large language model (GPT4) with a host biomarker (FABP4) to address an important clinical problem which is the difficulty of differentiating infection from other causes of respiratory failure in ICU patients. Biomarkers alone have always been intended to supplement not replace clinical judgement and the use of GPT4 attempts to codify clinical factors and improve the standard management of this difficult group. This is a particularly challenging group of patients since signs and symptoms so frequently overlap, however by virtue of their critical illness and a slim margin of error to be wrong, any test must have nearly 100% sensitivity if expected to be adopted.

The investigators show that the combination of GPT4 and FABP4 has greater accuracy than either method alone as well as greater accuracy than clinical diagnosis at the time of enrollment as compared to a gold standard of clinical adjudication which takes advantage of the entire hospital course and laboratory testing.

This study provides some proof of concept of this novel idea, but the study has number of limitations that restrict any definitive conclusions. The primary issue is that the tests, outcomes and gold standard are not clearly defined or seem to

measure different things. FABP4 is associated with LRTI whether viral or bacterial, GPT4 predicts LRTI – not clear if any infection or specifically bacterial LRTI. Clinician diagnosis of LRTI is measured by antibiotic use, so presumably bacterial LRTI and final adjudication is stated according to CDC PNEU1 criteria (bacterial) leaving the reviewer uncertain as to how COVID cases were classified since validation was done during the pandemic.

As a general suggestion it would be ideal to provide some explanations of methods for people not intimately familiar with this biomarker and GPT4 to make the paper more readable.

Introduction

Lines 66-77 please provide some explanation of what FABP4 is

Lines 78-79 – references 16-18 do not support the statement that LLMs have remarkable performance to assist clinical reasoning. Reference 16 concludes that Chat bot AI without expert medical advice can be hazardous.

Methods

Lines 181-182 – enrollment is noted to be within 72 hours of intubation. As I understand the study, this was the point the TA sample was collected for sequencing and serves as the reference point for judging clinician diagnosis, the notes and CXR results. Things change rapidly in the ICU, so would it be correct to assume if a person was enrolled at 72 hours of intubation, the notes and CXR would be from 1 day prior, not at the time of intubation or admission to the ICU? Lines 200-202 note the clinician's use of antibiotics on admission to the ICU, not at the time of enrollment. This potential variability would seem an issue in a study with a small sample size.

Lines 183-184 The derivation and validation cohorts were enrolled in very different times (pre-pandemic vs pandemic) with resulting significantly different LRTI causes which needs to be addressed.

Lines 200-203. One could question the validity of using antibiotic use at the time of admission to ICU as a surrogate for clinician team diagnosis of LRTI. Was one dose, 24 hours of antibiotics sufficient to call LRTI? If antibiotics were not prescribed but rather oseltamivir or remdesivir prescribed, was this considered LRTI?

The GPT4 and FABP4 are dichotomous – LRTI or not whereas in a clinical setting infectious process may be on the differential but not the leading diagnosis.

In addition, my understanding is the FABP4 discriminates infection of any cause – not specifically bacterial and is noted to be markedly depleted in SARS-CoV-2 infection. So, the biomarker is measuring any infection, not necessarily bacterial and thus, focusing on antibiotic use doesn't entirely make sense.

Lines 226 - 233. Would it be possible to explain some of the GPT4 methods in language more accessible to a non AI familiar reader? It is not intuitive that if given 3 tries the GPT4 would read the information differently.

Lines 245-246 – Using 5-fold in the derivation and 3-fold cross validation in the validation cohort should be noted as a limitation

Results

Lines 112- Why is the test only one sided? Is it not theoretically possible the combined measures could worsen accuracy?

Lines 127-142 The comparison of GPT to the 3 physicians given the same information as the Chat bot seems the fairest comparison and the discrepancies to identify potential biases is of interest. Given that CXRs are frequently read with ambiguity and as "cannot rule out infection", it might be interesting to provide more detail in this regard.

Figure 1- it notable that ½ the cases could not be definitively adjudicated. This is not surprising but does raise the issue of generalizability in an ICU setting and could be noted in the discussion.

Figure 2. Please place an explanation of the color red/green in the footnote

Supplement- it would be good to give examples of the GPT of non LRTI as well as LRTI

Discussion

Lines 148-154. I would suggest softening this language – stopping empiric antibiotics at the time of crisis based on the eventual diagnosis would be extremely difficult. In addition, there is no mention of potential harm to those with LRTI missed and not treated appropriately paying most attention to the 8/37 missed in the derivation cohort. It is hard to interpret the 0 misses during the pandemic when there were very few bacterial infections.

A more robust discussion of the limitations is needed- GPT given only one note when a provider has access to the entire hospital course, focusing on antibiotics as a measure of clinical diagnosis of LRTI (especially at the height of COVID), very different derivation and validation cohorts, the use of different fold cross validation in the derivation and validation cohorts.

Although the authors note that future work is needed, the study really should be seen as proof of concept.

(Remarks on code availability)

Version 1:

Reviewer comments:

Reviewer #1

(Remarks to the Author)

The authors have satisfactorily provided responses to the points I initially raised during my review. I see the paper now in a good stage that would merit publication. This paper deals with a very interesting and novel concept that has the potential to move the field forward.

(Remarks on code availability)

The repository is well documented and contain all the necessary steps for reproducing the study. Even the data the authors used is freely available on their repository.

Reviewer #2

(Remarks to the Author)

The authors have been very responsive to prior comments particularly defining antimicrobial use rather than antibiotics. I do have a few additional comments and suggestions.

The clarifications regarding a 72-hour window after intubation to enrollment and it is reassuring that the average time since intubation was 27-34 hours, but I still wonder if there are a few that were enrolled after 3 days of intubation that these individuals were intrinsically different. At the least, the authors should add days from intubation in Table 1 similar to what was done for the note and CXR. It might be worth looking at those with longer times ventilated prior to enrollment compared to short times to see if there are any differences in results.

It reflects my lack of understanding of AI, how a machine given the exact same input data would produce different results, but I appreciate the statement on line 302 as an explanation of why 3 sessions were used.

Figure 2. I would still explain the color not just the intensity. After "Intensity of color in the confusion matrices reflects the % of patients in each quadrant: Red color indicates the initial ICU diagnosis and green indicates the FABP4-GPT4 classifier.

Since 2-sided T tests were done and did not change the results, why not include a statement saying so after line 127.

(Remarks on code availability)

REVIEWER COMMENTS

Reviewer #1 (Remarks to the Author):

I read with interest the work from Van Phan et al. In their work, the authors proposed a diagnostic system for low respiratory tract infection. The system is based on large language models where the input is electronic medical record combined with measurements of the transcriptomic biomarker FABP4. While this is a remarkable and easy-to-read paper that bridges well the language of machine learning and medical sciences, a future iteration of the work might benefit from clarification of the following points:

R1Q1:

My biggest concern of the paper is regarding potential biases that might have been introduced in the EMRs that might have influenced GPT4's decision. The authors did not make clear the level of details of each EMR that was fed to GPT4. This needs to be considered as GPT models are sensitive to phasing, structure, and format of input data. What are the similarities between the EMRs? Without a clear standardization, GPT may perform better on some groups.

Response. We appreciate this point and have now better clarified the details of the EMR data provided to GPT4. Our goal was to provide a realistic scenario in which GPT4 could be used to support clinician decision making. To that end, we provided GPT4 with the data that a physician would have on the day of care: namely, a chest X-ray read from that day and the primary team note from the day prior. Notes varied in length, content, and structure according to the managing service (breakdown listed in Table 1), which reflects the variety of clinical practice and documentation within our health system. We have clarified this as follows:

Line 247: "Notes were written in the Epic EMR platform by physicians from the primary care team, which included Internal Medicine, Critical Care, and several other additional services (**Table 1**). Notes varied in length and structure, reflecting the real-world diversity of clinical practice, and allowing a realistic scenario for GPT-4 use."

R1Q2:

The authors used only logistic regression models to predict (FABP4, GPT4, and both). What is the rationale on using this model? Were other models tested?

Response. We appreciate this question. We chose logistic regression because among machine learning methods, it was best suited for the 1-2 features we sought to test within the sample size of the cohorts. More specifically, logistic regression is less vulnerable to overfitting compared to other more complex models such as a random forest or gradient boosting classifiers. In addition, logistic regression is among the most broadly utilized statistical methods reported in the medical literature, and we believed that this inherent familiarity and interpretability would be particularly appealing in clinical settings compared to more advanced but less transparent machine learning models.

We have now clarified this in the manuscript as follows:

Methods, Line 268: We chose to use logistic regression because among machine learning methods, it was best suited for the 1-2 features we sought to test within the sample size of the cohorts. More specifically, logistic regression is less vulnerable to overfitting compared to other more complex models such as a random forest or gradient boosting classifiers. In addition, logistic regression is among the most broadly utilized

statistical methods reported in the medical literature, and we believed that this inherent familiarity and interpretability would be particularly appealing in clinical settings compared to more advanced but less transparent machine learning models.

R1Q3:

In lines 115-117, the authors bring up the classification metrics of their combined classifier. The authors could have used different metrics that consider the imbalanced nature of their adjudication data.

Response. We thank the reviewer for this suggestion. While our data set was not highly imbalanced (the proportion of LRTI patients was $37/86 = 43\%$), we do acknowledge the importance of such imbalance-focused metrics. We have added the positive predictive value, negative predictive value and F1 score:

Line 132: "... demonstrated an average sensitivity of 86%, specificity of 98%, accuracy of 93%, positive predictive value of 97%, negative predictive value of 91%, and F1 score of 0.91."

R1Q4:

A clear benefit demonstrated by the authors in using such system was its ability to classify true negatives (where the medical team incorrectly classified ~70% of the true negatives as false positives). The performance in the automated system and the medical team to classify true positives was similar (96% versus 100%, respectively). In order to better justify the use of the proposed system, I suggest the authors to add insights on how long it took for the medical team to make the diagnosis of the true positive classes.

Response: We appreciate the reviewer's point and have now assessed time to microbiologic diagnosis for the true positive cases. Based on this, we found that the medical team would have been able to make a microbiologically confirmed LRTI diagnosis in a median of 56.3 hours from intubation in the derivation cohort, and in a median of 59.6 hours for patients in the validation cohort who had diagnoses other than COVID-19. Due to the incorporation of universal SARS-CoV-2 testing prior to hospital admission during the period of validation cohort enrollment, most patients with COVID-19 were diagnosed prior to intubation in the validation cohort. We have addressed this further in the results:

Line 99: "Time to microbiologic diagnosis from intubation was a median of 56.3 hours (interquartile range (IQR) 28.9 – 73.4 hours) in the derivation cohort and 59.6 (33.9 – 88.8 hours) in the validation cohort for patients with diagnoses other than COVID-19. Due to the incorporation of universal SARS-CoV-2 testing prior to hospital admission during the period of validation cohort enrollment, most patients with COVID-19 were diagnosed prior to intubation in the validation cohort."

Reviewer #1 (Remarks on code availability):

The code and data are publicly available. The code is well documented and easy to reproduce.

Response. We thank the reviewer for reviewing our code and appreciate that they found it well documented and easy to reproduce.

Reviewer #2 (Remarks to the Author):

This is an interesting study that attempts to combine a large language model (GPT4) with a host biomarker (FABP4) to address an important clinical problem which is the difficulty of differentiating infection from other causes of respiratory failure in ICU patients. Biomarkers alone have always been intended to supplement not replace clinical judgement and the use of GPT4 attempts to codify clinical factors and improve the standard management of this difficult group. This is a particularly challenging group of patients since signs and symptoms so frequently overlap, however by virtue of their critical illness and a slim margin of error to be wrong, any test must have nearly 100% sensitivity if expected to be adopted. The investigators show that the combination of GPT4 and FABP4 has greater accuracy than either method alone as well as greater accuracy than clinical diagnosis at the time of enrollment as compared to a gold standard of clinical adjudication which takes advantage of the entire hospital course and laboratory testing.

R2Q1:

This study provides some proof of concept of this novel idea, but the study has number of limitations that restrict any definitive conclusions. The primary issue is that the tests, outcomes and gold standard are not clearly defined or seem to measure different things. FABP4 is associated with LRTI whether viral or bacterial, GPT4 predicts LRTI – not clear if any infection or specifically bacterial LRTI. Clinician diagnosis of LRTI is measured by antibiotic use, so presumably bacterial LRTI and final adjudication is stated according to CDC PNEU1 criteria (bacterial) leaving the reviewer uncertain as to how COVID cases were classified since validation was done during the pandemic.

Response

First, we would like to acknowledge the reviewer's important point that much work in the field of lower respiratory tract infections has focused exclusively on bacterial pathogens, although data suggest viral etiologies may be the most common (Jain et al. 2015) with this especially true during the COVID-19 pandemic.

Second, we appreciate the opportunity to clarify that LRTI was defined agnostic to microbiological etiology (bacterial, viral, or fungal). Gold standard multi-physician adjudication of patient LRTI status was performed using a combination of the US CDC PNEU1 clinical criteria, which are agnostic to LRTI etiology (<https://www.cdc.gov/nhsn/pdfs/checklists/pneu-checklist-508.pdf>). In addition to these clinical criteria, we also required detection of an established respiratory pathogen by standard of care in-hospital testing, to provide the greatest confidence in accurate true positive LRTI assignments.

Third, we would also like to clarify that we used both GPT4 and *FABP4* to assess all-cause LRTI; our GPT4 prompt deliberately does not specify cause of infection.

Finally, we would like to clarify that initial clinician diagnosis of LRTI was measured by antimicrobial use, including antibacterials, antivirals, and/or antifungals. For instance, remdesivir was part of our hospital's clinical guidelines for treatment of patients with COVID-19 during the period of study enrollment; and oseltamivir was used empirically for patients with suspected or proven influenza.

To clarify these important points, we have amended the text as follows

Line 254: “The clinical treatment team’s LRTI diagnosis was extrapolated based on administration of antimicrobials (antibacterial, antiviral, and/or antifungal agents) for at least 24 hours of empiric treatment of LRTI within one day of study enrollment, excluding agents given for established non-pulmonary infections or prophylaxis.”

Line 65: “*FABP4*’s consistent performance across cohorts with differing microbiology – predominantly bacterial infections in adults and viral in children – suggested that *FABP4* is agnostic to the type of pathogen causing LRTI.⁸”

Line 96: “The majority of LRTI patients in the derivation cohort had bacterial etiologies of LRTI, while the validation cohort, which was largely recruited during the COVID-19 pandemic, was predominantly viral (**Table 1**).”

Line 116: “All 24 of the derivation cohort No LRTI patients unnecessarily treated for LRTI received antibacterial coverage, with four additionally receiving empiric therapy directed at viral and/or fungal pathogens (**Supplementary Table 1**).”

Line 135: “Next, we assessed the validation cohort (**Figure 1**), in which the primary medical team correctly identified 25/25 (100%) of LRTI cases but unnecessarily treated with antibacterials in 7/22 (32%) of patients in the No LRTI group, equating to an accuracy of 85% (**Figure 2C, Supplementary Table 1**).”

Additionally, we have added a new Supplementary Table 1, which details the classes of antimicrobials received by No LRTI patients.

R2Q2:

As a general suggestion it would be ideal to provide some explanations of methods for people not intimately familiar with this biomarker and GPT4 to make the paper more readable.

Response: We appreciate this suggestion and have added further detail below in response to the Reviewer’s suggestions. More specifically, we have revised the introduction to provide additional context on GPT-4 in an effort to improve accessibility for readers who may be less familiar with large language model-based artificial intelligence:

Line 76: “Large language models (LLMs) such as Generative Pre-trained Transformer 4 (GPT-4) represent a new class of artificial intelligence tools with potential utility across a diversity of medical applications¹⁴. GPT-4 provides a text interface in which a clinician or other user may pose questions, to which GPT-4 then responds in conversational language. While LLMs have demonstrated remarkable performance for some medical applications, including image interpretation¹⁵⁻¹⁷, patient risk stratification¹⁸, their utility for aiding clinical reasoning has remained unclear¹⁹⁻²¹, and their utility for diagnosing LRTI or other critical illness syndromes based on electronic medical record (EMR) data has not been assessed. Here, we address this gap by building a diagnostic classifier combining *FABP4* with GPT-4 analysis of EMR data. We find that this combination affords remarkably accurate LRTI diagnosis, suggesting a promising new approach to improve the care of critically ill patients.”

R2Q3:

Lines 66-77 please provide some explanation of what *FABP4* is

Response: We thank the reviewer for this point and have now provided additional background on *FABP4* as follows:

Line 65: “*FABP4*’s consistent performance across cohorts with differing microbiology – predominantly bacterial infections in adults and viral in children – suggested that *FABP4* is agnostic to the type of pathogen causing LRTI⁸. Mechanistically, fatty acid binding proteins are a family of intracellular proteins that modulate fatty acid trafficking and inflammatory signaling⁹. *FABP4* specifically is highly expressed in tissue-resident alveolar macrophages which are preferentially depleted during respiratory infections including bacterial pneumonia¹⁰ and COVID-19¹¹.”

R2Q4:

Lines 78-79 – references 16-18 do not support the statement that LLMs have remarkable performance to assist clinical reasoning. Reference 16 concludes that Chat bot AI without expert medical advice can be hazardous.

Response: We appreciate this point and the opportunity to correct this oversight. We have revised as follows:

Line 80: “While LLMs have demonstrated remarkable performance for some medical applications, including image interpretation¹⁵⁻¹⁷, patient risk stratification¹⁸, their utility for aiding clinical reasoning has remained unclear¹⁹⁻²¹, and their utility for diagnosing LRTI or other critical illness syndromes based on electronic medical record (EMR) data has not been assessed”

Methods

R2Q5:

Lines 181-182 – enrollment is noted to be within 72 hours of intubation. As I understand the study, this was the point the TA sample was collected for sequencing and serves as the reference point for judging clinician diagnosis, the notes and CXR results. Things change rapidly in the ICU, so would it be correct to assume if a person was enrolled at 72 hours of intubation, the notes and CXR would be from 1 day prior, not at the time of intubation or admission to the ICU? Lines 200-202 note the clinician’s use of antibiotics on admission to the ICU, not at the time of enrollment. This potential variability would seem an issue in a study with a small sample size.

Response: We appreciate the opportunity to provide clarification regarding the timing of events.

Study enrollment occurred a median of 26.9 (95% CI 27.6-36.0) hours after intubation in the derivation cohort, and 33.8 (95% CI 14.3-56.4) hours after intubation in the validation cohort. We centered analyses at the time of enrollment, which was close to time of intubation, largely because that is when the TA sample was collected, and because it still represented a clinically relevant timepoint at which LRTI status was still uncertain for most patients.

CXR reads used for the GPT4 model were obtained from the day of study enrollment, or as close to the day of enrollment as possible. Notes were obtained from the day prior to enrollment, or as close to that day as possible. Days from enrollment of notes and CXR are shown in Table 1.

We also would like to clarify that the clinical team’s diagnosis of LRTI was based on antimicrobial receipt for at least 24 hours within one day of study enrollment (also discussed in R2Q7).

We realize that our initial phrasing was confusing and have amended it as follows:

Line 254: “The clinical treatment team’s LRTI diagnosis was extrapolated based on administration of empiric antimicrobials (antibacterial, antiviral, and/or antifungal agents) for at least 24 hours within one day of study enrollment, excluding agents given for established non-pulmonary infections or prophylaxis.”

R2Q6:

Lines 183-184 The derivation and validation cohorts were enrolled in very different times (pre-pandemic vs pandemic) with resulting significantly different LRTI causes which needs to be addressed.

Response: We concur with the reviewer’s point, and now better address this as follows:

Line 202: “Moreover, this approach yielded promising results in two cohorts of patients with very different microbial etiologies of LRTI, suggesting that both *FABP4* and LLM analysis of EMR data may have utility as diagnostic approaches, agnostic to type of LRTI pathogen. Bacterial LRTI was predominant in the pre-pandemic derivation cohort while the validation cohort, which was enrolling during the COVID-19 pandemic, primarily consisted of patients with viral LRTI.”

R2Q7:

Lines 200-203. One could question the validity of using antibiotic use at the time of admission to ICU as a surrogate for clinician team diagnosis of LRTI. Was one dose, 24 hours of antibiotics sufficient to call LRTI? If antibiotics were not prescribed but rather oseltamivir or remdesivir prescribed, was this considered LRTI?

Response: We appreciate this point, especially given that our validation cohort was enrolled during years in which the predominating pathogen causing LRTI was viral (SARS-CoV-2). First, we would like to clarify that we defined antibiotic use as antimicrobials directed at any pathogen class (viral, fungal, or bacterial) intended for LRTI treatment, unless the agent was being already administered for an established non-pulmonary infection or given as a prophylactic agent. Thus, we would have considered receipt of antiviral agents alone to indicate a clinician diagnosis of LRTI. We also now use the more correct term ‘antimicrobial’ throughout.

Second, we would like to clarify that the clinical team’s diagnosis of LRTI was based on antimicrobial receipt for at least 24 hours within one day of study enrollment.

We have sought to clarify this in the text as follows:

Line 254: “The clinical treatment team’s LRTI diagnosis was extrapolated based on administration of empiric antimicrobials (antibacterial, antiviral, and/or antifungal agents) for at least 24 hours within one day of study enrollment, excluding agents given for established non-pulmonary infections or prophylaxis.”

The Reviewer’s point also prompted us to re-review all of our antimicrobial use data, and in doing so we recognized that 2 patients in the derivation cohort were miscategorized, both of whom received antimicrobials on the day prior to enrollment and who had been previously incorrectly assigned a clinical team diagnosis of “No LRTI” as opposed to “LRTI”. We apologize for this oversight which we have corrected. Importantly, the central findings and conclusions of our study

remain the same. This correction is reflected in the first panel of Figure 2, in Table 1, and in the text as follows:

Line 113: The medical team correctly identified 37/37 (100%) of true LRTI cases but incorrectly called LRTI in 24/49 (49%) of patients in the No LRTI group, equating to an accuracy of 72% (**Figure 2A, Table 1**). All 24 of the derivation cohort No LRTI patients unnecessarily treated for LRTI received antibacterial coverage, with four additionally receiving empiric therapy directed at viral and/or fungal pathogens (**Supplementary Table 1**).

Line 173: Had our integrated classifier results been theoretically available at time of ICU admission, we estimate that inappropriate antimicrobial use might have been prevented in 20/24 (83%) and 7/7 (100%) of No LRTI patients who were unnecessarily treated in the derivation and validation cohorts, respectively.

The GPT4 and FABP4 are dichotomous – LRTI or not whereas in a clinical setting infectious process may be on the differential but not the leading diagnosis.

Response: We appreciate this point, which we have now emphasized as follows:

Line 183: “...many patients with clinical LRTI never have a confirmed microbiologic diagnosis; across the study cohorts, approximately 50% of patients were clinically adjudicated as having LRTI without an identified microbial pathogen, or of having an uncertain LRTI status (Figure 1). Although we focused on the unequivocal cases of proven LRTI or No LRTI to develop and test our GPT-4 and *FABP4*/GPT-4 classifiers, it is those cases without a clear diagnosis, in which LRTI is considered as one possible diagnosis among many, where this method may ultimately prove most useful. A future randomized clinical trial will be needed to conclusively test this.

R2Q8:

In addition, my understanding is the FABP4 discriminates infection of any cause – not specifically bacterial and is noted to be markedly depleted in SARS-CoV-2 infection. So, the biomarker is measuring any infection, not necessarily bacterial and thus, focusing on antibiotic use doesn't entirely make sense.

Response: As noted above in R2Q7, we would like to clarify that we defined antibiotic use as antimicrobials directed at any pathogen class (viral, fungal, or bacterial) intended for LRTI treatment, unless the agent was being already administered for an established non-pulmonary infection or given as a prophylactic agent. Thus, we would have considered receipt of antiviral agents alone to indicate a clinician diagnosis of LRTI. We now use the more correct term ‘antimicrobial’.

Line 254: “The clinical treatment team’s ICU admission LRTI diagnosis was extrapolated based on administration of antimicrobials (antibacterial, antiviral, and/or antifungal agents) for empiric treatment of LRTI on the day of study enrollment or the subsequent day, excluding agents being given for established non-pulmonary infections or prophylaxis.”

We would like to also note that in our derivation cohort, bacterial causes of LRTI predominated; and in our validation cohort, SARS-CoV-2 was the leading cause of infection. Both bacterial causes of LRTI and SARS-CoV-2 LRTI had treatments available at time of patient enrollment and

empiric antibacterial and/or remdesivir therapy would be expected to have been initiated prior to confirmation of diagnosis. Additionally, as per IDSA guidelines, antiviral treatment for patients with possible influenza A would be initiated in our hospitals prior to confirmation of diagnosis.

Finally, we would like to agree with the reviewer that antimicrobial administration is an imperfect metric to extrapolate clinical team diagnosis of LRTI. But given that clinician documentation can be complex, fragmented, and subject to different interpretations, we believe that antimicrobial use was one of the most objective and unbiased options for retrospectively estimating the clinical team's diagnosis.

R2Q9:

Lines 226 - 233. Would it be possible to explain some of the GPT4 methods in language more accessible to a non AI familiar reader? It is not intuitive that if given 3 tries the GPT4 would read the information differently.

Response: We thank the reviewer for this point and have now added the following to make the GPT4 methods more accessible:

Line 76: "Large language models (LLMs) such as Generative Pre-trained Transformer 4 (GPT-4) represent a new class of artificial intelligence tools with potential utility across a diversity of medical applications(14). GPT-4 provides a text interface in which a clinician or other user may pose questions, to which GPT-4 then responds in conversational language. While LLMs have demonstrated remarkable performance for some medical applications, including image interpretation(15-17), patient risk stratification(18), their utility for aiding clinical reasoning has remained unclear(19-21) and their utility for diagnosing LRTI or other critical illness syndromes based on electronic medical record (EMR) data has not been assessed. In this study, we considered the question of whether GPT-4 could be used to boost the performance of host biomarker-based LRTI diagnosis. Here, we address this gap by building a diagnostic classifier combining *FABP4* with GPT-4 analysis of EMR data. We find that this combination affords remarkably accurate LRTI diagnosis, suggesting a promising new approach to improve the care of critically ill patients."

Line 302: "We found that GPT-4 would sometimes give different answers to the same prompt and EMR input data in separate chat sessions. Therefore, for each patient, GPT-4 was asked to diagnose LRTI in three separate sessions. A per-patient GPT-4 score was calculated based on the total number of LRTI-positive diagnoses made by GPT-4."

R2Q10:

Lines 245-246 – Using 5-fold in the derivation and 3-fold cross validation in the validation cohort should be noted as a limitation

Response. We agree with the reviewer, and have modified the text as follows:

Line 209: "Limitations of this study include a relatively small sample size, particularly in the validation cohort, which necessitated the use of 3-fold (versus 5-fold) cross-validation."

Results

R2Q11:

Lines 112- Why is the test only one sided? Is it not theoretically possible the combined measures could worsen accuracy?

Response. It is indeed possible that the combined classifier could perform worse. Our primary objective was to test if the combination of the biomarker and GPT-4 *improves* the classifier's performance. Therefore, the one-sided test was more appropriate. A decrease in performance would result in a non-significant P-value, which aligns with our hypothesis. Prompted by the reviewer's question, we also applied 2-sided T-tests and found that our results, in terms of statistical significance, and conclusions, did not change.

We have added the following to the manuscript to clarify our rationale for the one-sided test:

Line 127: "We used the one-sided paired t-test because we were primarily interested in whether the integrated classifier outperformed each individual classifier."

R2Q12:

Lines 127-142 The comparison of GPT to the 3 physicians given the same information as the Chat bot seems the fairest comparison and the discrepancies to identify potential biases is of interest. Given that CXRs are frequently read with ambiguity and as "cannot rule out infection", it might be interesting to provide more detail in this regard.

Response: We appreciate the reviewer's point and have now added further detail in the manuscript and supplement regarding these cases.

Line 154: "Of the nine patients more frequently diagnosed with LRTI by GPT-4 versus the comparison physicians (**Figure 3B**), six had clinical notes with no mention of LRTI, but explicit concern for LRTI in the CXR report, as judged by mention of "pneumonia" and/or "infection" in the radiologist read (CXR reports provided in **Supplementary Appendix 4**)."

In addition, we have provided the full text of the CXR reads of the 9 patients assessed by GPT-4 to have LRTI, but by comparison physicians to not have LRTI, in a new Appendix 4.

R2Q13:

Figure 1- it notable that ½ the cases could not be definitively adjudicated. This is not surprising but does raise the issue of generalizability in an ICU setting and could be noted in the discussion.

Response: It is indeed notable and not surprising based on existing literature demonstrating that the majority of pneumonia/LRTI cases in hospitalized patients never have a microbe identified (e.g., Jain et al. NEJM. 2015). We appreciate the need to raise the issue of generalizability of our results in ICU settings and have added the following:

Line 221: "Future work can test whether GPT-4 can improve the marginal performance of widely available clinical biomarkers such as C-reactive protein, assess the generalizability of FABP4/GPT-4 classifier performance in larger independent cohorts of ICU patients..."

We also appreciate the opportunity to clarify that we focused exclusively on patients with the most unambiguous LRTI status to provide the most confident and accurate ground truth for training and validating our classifiers. We absolutely recognize that cases without an identified microbe or a clear clinical diagnosis may in fact be the ones for which our classifier may be most useful. We also recognize that conclusively determining that would require a randomized controlled trial.

Line 183: “Finally, many patients with clinical LRTI never have a confirmed microbiologic diagnosis²; across the study cohorts, approximately 50% of patients were clinically adjudicated as having LRTI without an identified microbial pathogen, or of having an uncertain LRTI status (Figure 1). Although we focused on the unequivocal cases of proven LRTI or No LRTI to develop and test our GPT-4 and *FABP4*/GPT-4 classifiers, it is those cases without a clear diagnosis, in which LRTI is considered as one possible diagnosis among many, where this method may ultimately prove most useful. A future randomized clinical trial will be needed to conclusively test this.”

R2Q14:

Figure 2. Please place an explanation of the color red/green in the footnote

We appreciate this point and have amended the legend as follows:

Line 444: Figure 2. Performance of *FABP4*, GPT-4 and integrated LRTI diagnostic classifiers in the derivation and validation cohorts. **A)** Confusion matrices for initial ICU diagnosis and the integrated *FABP4*/GPT-4 classifier in the derivation cohort. **B)** Receiver operating characteristic curves from GPT-4 classifier, *FABP4* classifier, and integrated *FABP4*/GPT-4 classifier in the derivation cohort. **C)** Confusion matrices for initial ICU diagnosis and the integrated *FABP4*/GPT-4 classifier in the validation cohort. **D)** Receiver operating characteristic curves from GPT-4 classifier, *FABP4* classifier, and integrated *FABP4*/GPT-4 classifier in the validation cohort. In panels A and C, the classifiers output an LRTI diagnosis if the patients had a predicted out-of-fold LRTI probability of 50% or higher, Intensity of color in confusion matrices reflects percentage of patients in each quadrant. In panels B and D, the area under the curves (AUCs) are presented as mean \pm standard deviation.

R2Q15:

Supplement- it would be good to give examples of the GPT of non LRTI as well as LRTI

Response: We appreciate this point and have included a new Appendix 3 with an example of a GPT-4 No LRTI diagnosis. We have also added the following to the text:

Line 297: “The validation cohort included patients enrolled during the height of the COVID-19 pandemic and thus we redacted the terms “SARS-CoV-2” or “COVID-19” from their notes to avoid biasing the GPT-4 analysis. In our final version of the prompt (**Supplementary Appendix 1**), we asked GPT-4 to choose either LRTI or no LRTI, as exemplified in (**Supplementary Appendix 2 and 3**).”

Discussion

R2Q16

Lines 148-154. I would suggest softening this language – stopping empiric antibiotics at the time of crisis based on the eventual diagnosis would be extremely difficult. In addition, there is no mention of potential harm to those with LRTI missed and not treated appropriately paying most attention to the 8/37 missed in the derivation cohort. It is hard to interpret the 0 misses during the pandemic when there were very few bacterial infections.

Response: We respect and understand this point and have revised the discussion as follows:

Line 179: “However, given the challenges of de-escalating of antimicrobials in critically ill patients, and the potential consequences of inappropriately stopping treatment in a patient with true LRTI, our results serve primarily as a proof of concept requiring further validation.”

R2Q17

A more robust discussion of the limitations is needed- GPT given only one note when a provider has access to the entire hospital course, focusing on antibiotics as a measure of clinical diagnosis of LRTI (especially at the height of COVID), very different derivation and validation cohorts, the use of different fold cross validation in the derivation and validation cohorts.

Although the authors note that future work is needed, the study really should be seen as proof of concept.

Response: We appreciate these points and have revised the discussion as follows:

Line 209: “Limitations of this study include a relatively small sample size, particularly in the validation cohort, which necessitated the use of 3-fold (versus 5-fold) cross-validation. In addition, our focus on mechanically ventilated patients may limit generalizability to less severe respiratory illnesses. Antimicrobial administration is an imperfect proxy for clinical team LRTI diagnosis; however, it was an objective, reproducible and unbiased option for retrospectively estimating the clinical team’s decision making. We restricted GPT-4 analyses to a single EMR note and CXR read, and it is possible that assessment of more complete EMR data would have led to improved, or different, performance. Given these limitations, this study is best seen as a proof of concept that establishes the feasibility and promise of a diagnostic approach that combines artificial intelligence-based EMR analysis with a host biomarker.”

Reviewer #1 (Remarks to the Author)

The authors have satisfactorily provided responses to the points I initially raised during my review. I see the paper now in a good stage that would merit publication. This paper deals with a very interesting and novel concept that has the potential to move the field forward.

(Remarks on code availability)

The repository is well documented and contain all the necessary steps for reproducing the study. Even the data the authors used is freely available on their repository.

Response: We appreciate the Reviewer's feedback.

Reviewer #2 (Remarks to the Author)

The authors have been very responsive to prior comments particularly defining antimicrobial use rather than antibiotics. I do have a few additional comments and suggestions.

The clarifications regarding a 72-hour window after intubation to enrollment and it is reassuring that the average time since intubation was 27-34 hours, but I still wonder if there are a few that were enrolled after 3 days of intubation that these individuals were intrinsically different. At the least, the authors should add days from intubation in Table 1 similar to what was done for the note and CXR. It might be worth looking at those with longer times ventilated prior to enrollment compared to short times to see if there are any differences in results.

Response: We appreciate this feedback. To provide further granular detail, we have now added time in hours from intubation to sampling, which occurred soon after enrollment, to Table 1. Only four patients in the derivation cohort and two in the validation cohort had times that exceeded 72 hours from intubation to sample collection. We did not identify any differences in these patients based on the metrics outlined in Table 1, however given the small number of such cases, we could not meaningfully test for differences between them and other participants.

It reflects my lack of understanding of AI, how a machine given the exact same input data would produce different results, but I appreciate the statement on line 302 as an explanation of why 3 sessions were used.

Response: We are glad that this explanation has been helpful in understanding the AI outputs described in our study.

Figure 2. I would still explain the color not just the intensity. After "Intensity of color in the confusion matrices reflects the % of patients in each quadrant: Red color indicates the initial ICU diagnosis and green indicates the FABP4-GPT4 classifier.

Response: We appreciate this suggestion and have adjusted the legend as follows:

Figure 2. Performance of FABP4, GPT-4 and integrated LRTI diagnostic classifiers in the derivation and validation cohorts. A) Confusion matrices for initial ICU diagnosis and the integrated FABP4/GPT-4 classifier in the derivation cohort. **B)** Receiver operating characteristic curves from GPT-4 classifier, FABP4 classifier, and integrated FABP4/GPT-4 classifier in the derivation cohort. **C)** Confusion matrices for initial ICU diagnosis and the integrated FABP4/GPT-4 classifier in the validation cohort. **D)** Receiver operating characteristic curves from GPT-4 classifier, FABP4 classifier, and integrated FABP4/GPT-4 classifier in the validation cohort. In

panels A and C, the classifiers output an LRTI diagnosis if the patients had a predicted out-of-fold LRTI probability of 50% or higher, Intensity of color in confusion matrices reflects percentage of patients in each quadrant; red indicates the initial ICU diagnosis and green is the integrated *FABP4*/*GPT-4* classifier. In panels B and D, the area under the curves (AUCs) are presented as mean \pm standard deviation.

Since 2-sided T tests were done and did not change the results, why not include a statement saying so after line 127.

Response: We appreciate the Reviewer's point and have added the following detail to the text:

Line 126: "Combining *FABP4* and *GPT-4* into a single logistic regression model achieved an AUC of 0.93 ± 0.08 (**Figure 2B**), outperforming both *FABP4* ($P = 0.002$, one-sided paired t-test) and *GPT-4* alone ($P = 0.008$, one-sided paired t-test). We used the one-sided paired t-test because we were primarily interested in whether the integrated classifier outperformed each individual classifier, but the difference remained statistically significant when using two-sided t-tests ($P = 0.004$ and $P = 0.016$, respectively)."